# A Comparison of Rhizospheric and Endophytic Bacteria in Early and Late-Maturing Pumpkin Varieties

**DOI:** 10.3390/microorganisms10081667

**Published:** 2022-08-18

**Authors:** Siyu Chen, Renliu Qin, Da Yang, Wenjun Liu, Shangdong Yang

**Affiliations:** 1Guangxi Key Laboratory of Agro-Environment and Agro-Products Safety, National Demonstration Center for Experimental Plant Science Education, Agricultural College, Guangxi University, Nanning 530004, China; 2Vegetable Research Institute, Guangxi Academy of Agricultural Sciences, Nanning 530007, China

**Keywords:** pumpkin (*Cucurbita moschata* Duchesne), rhizosphere, endophytic bacterial compositions, soil fertility, high-throughput sequencing

## Abstract

To determine whether rhizospheric and endophytic bacteria contribute to the ripening of pumpkins, an analysis was conducted on rhizospheric and endophytic bacteria and soil fertility in the rhizospheres of early and late-maturing pumpkin varieties. The results showed higher nitrogen and abscisic acid content and more gibberellin-producing bacteria in the rhizospheres or endophytes of the early maturing varieties. Greater soil fertility and more abundant rhizospheric and endophytic bacterial genera with a greater metabolic function might be important mechanisms for early ripening. *Rhodococcus*, *Bacillus*, and *Arthrobacter* can be considered the functional bacteria in promoting pumpkin maturation. On the other hand, *Ralstonia* could be the functional bacterium that delays ripening.

## 1. Introduction

Pumpkin (*Cucurbita moschata* Duch.) is one of China’s main agricultural crops and is planted over a large area because of its strong adaptability and high nutritional profile [1]. At present, there are 27 cultivated and wild species that are not only different in color, shape, and size, but are also divided into early and late-maturity varieties [2].

However, an interesting phenomenon is that even though both varieties are planted at the same time and subjected to identical management, the early maturing varieties enter the flowering stage more quickly, whereas the late-maturing varieties have a longer growth period and enter the flowering stage later. Previous studies have shown that the growth and flowering of pumpkins are closely related to light [3], temperature [4], endogenous hormones [5], carbon and nitrogen metabolism, and dry matter accumulation [6]. In particular, temperature and light duration are the most important environmental factors because they affect dry matter accumulation and the synthesis and transformation of sugar and nitrogen [7].

Plant growth is closely related to hormone regulation, and auxin promotes fruit setting and growth by cooperating with cytokinin, gibberellin, and other hormones [8]. Ethylene [9,10] promotes stem elongation and fruit expansion [11], and together with salicylic [12] and abscisic acid [13,14,15], plays an important role in ripening.

It is well known that microbes in the soil, particularly the rhizospheres, closely interact with plants to produce phytohormones that affect the endogenous hormones [16]: auxin [17], ethylene [18,19], abscisic acid [20], jasmonic acid, salicylic acid [21], and cytokinin [22]. These affect plant growth directly or indirectly by influencing the rhizosphere environment [23]. Studies have also found that endophytic bacteria have interdependent relationships with auxin [24], ethylene [25], cytokinin [26], abscisic acid [27], and gibberellin [28].

Although early and late-maturing properties are mainly determined by the pumpkin variety and its corresponding ecological environment, the soil microbial community in the rhizosphere also affects growth and development [29,30]. Moreover, different plant genotypes affect the community characteristics and the diversity of endophytic microorganisms [31,32,33].

It is well-known that nutrients and the diversity of the rhizospheric and endophytic bacteria community provide the basis for vegetative and reproductive growth [34,35]. Of these, the importance of microorganisms is second only to climatic factors [36]. In addition, plant growth and development are related mainly to soil type and bacterial diversity. Additionally, enzymes are involved in fruit ripening and carbon cycling (e.g., glucosidase), nitrogen cycling (e.g., aminopeptidase), and phosphorous cycling (e.g., phosphatase) [37,38,39,40,41,42]. Soil enzyme activity is a sensitive index for detecting the soil–matter cycle and energy movement, and it can also control oxidation and the electron transfer of organic components in the formation of microbial energy [39]. Microbial diversity is not only beneficial for the mineralization and accumulation of organic matter, but it aids in the availability of soil nutrients, nutrient-cycling capacity, and growth. In the plant microhabitat, endophytes are important for the growth and health of the host plant [43] because endophytic bacteria increase the supply of nutrients and regulate hormone levels [44].

At present, studies on crop ripening have focused on cultivation factors [45], sowing dates [46], humidity [47], biotic or abiotic stress resistance [48], and different fertilizer ratios [49]. Minimal information is available on how different maturation stages regulate the rhizospheric and endophytic microbial community structure. Our aim is to elucidate differences in the soil fertility, rhizospheric bacterial community structure, and endophytic microorganisms between early and late-ripening pumpkin varieties—i.e., whether rhizospheric and endophytic bacteria contribute to maturation—and how the rhizospheric and endophytic bacterial community structure can be used as a novel tool for evaluating early or late ripening. Furthermore, bacteria that promote or delay maturation can be screened from the roots of both pumpkin varieties.

## 2. Materials and Methods

### 2.1. Field Site Description

The experiment was carried out at the LiJian Scientific Experimental Base of the Guangxi Academy of Agricultural Sciences (108°17′ E; 23°25′ N). The soil is quaternary laterite with 2–3% soil organic matter content and a pH value of 5.46. The physical and chemical characteristics of the soil are as follows: organic matter, 12.7 g kg^−1^; total nitrogen, 0.81 g kg^−1^; phosphorus, 0.39 g kg^−1^; potassium, 2.68 g kg^−1^; available nitrogen, 53.6 mg kg^−1^; available phosphorus, 9.0 mg kg^−1^; and available potassium, 88.9 mg kg^−1^.

### 2.2. Test Materials

Pumpkin varieties with two different maturities were used: three early maturing ((a) G1510-252-3-6, (b) G1681-2, and (c) G1536-169-3-11) and three late-maturing ((d) G1601-3-2, (e) G1713-2, and (f) G1215-1) varieties, all provided by the Vegetable Research Institute, Guangxi Academy of Agricultural Sciences (Figure 1). They were simultaneously sown and grown in the same field in February 2020 under identical management in a randomized block design with three replications. Through long-term field investigation, we found that the early maturing varieties all blossomed 10–15 days earlier than the late-maturing varieties. Soil samples from identical fields without any plant growth were also collected to determine the background (CK) levels.

### 2.3. Soil Sample Collection

Rhizospheric soil and plant samples of the different pumpkin varieties were collected on 20 June 2020 during the ripening period of the pumpkin fruit, using the shaking-off method as described by Riley and Barber [50]: a shovel was disinfected with a spray of 75% ethanol and then used to dig 50 cm around the roots in a radius of about 30 cm to loosen the soil, after which whole roots with soil were pulled up. After the bulk soil was shaken off, rhizospheric soil samples were carefully collected from the roots, put into sterile plastic bags, and packed with ice in a polystyrene foam box. Soil samples from the same field under identical management but with no pumpkins were collected for background (CK) data. Stem samples were collected separately from each plant using sterilized scissors, rinsed with sterile water to remove soil and appendages, and blotted dry with sterile filter paper. They were then placed in labeled, sealed sterile bags and immediately transferred to the laboratory, where they were sieved through a 2 mm stainless-steel mesh and stored in a refrigerator at 4 °C for immediate analysis or at −80 °C for later use.

### 2.4. Soil Physicochemical and Biological Properties

*β*-Glucosidase activity was determined using the chloroform fumigation-extraction method [51]. Aminopeptidase activity was determined using the colorimetric ninhydrin [52]. Acid phosphatase activity in soils was determined using the chloroform fumigation-extraction method [53].

### 2.5. Soil Bacterial DNA Extraction and Illumina Sequencing

The extraction, PCR amplification, and sequencing of total DNA from stem and soil samples were all completed by Majorbio Bio-Pharm Technology Co., Ltd. (Shanghai, China). Total DNA was extracted according to the instructions of the E.Z.N.A. DNA Kit (Omega Company, Norwalk, CT, USA). DNA concentration and purity were detected by a NanoDrop 2000 spectrophotometer (Thermo Company, Waltham, NJ, USA), and the purity and quality of the genomic DNA were checked on 1% agositol gel. The V3–4 hypervariable region of the bacterial 16S rRNA gene was amplified with the primers 338F (5′-ACT CCT ACG GGA GGC AGC AG-3′) and 806R (5′-GGA CTA CHVGGG TWT CTAAT-3′) using rhizospheric bacterial DNA as a template. Primers 799F and 1192R were selected for the first round of PCR amplification of the V5–V7 variable region, and primers 799F (5′-AACMGGA TTA GAT ACC CKG-3) and 1193R (5′-ACG TCA TCC CCA CCT TCC-3′) [54], using endophytic bacterial DNA as a template, were selected for the second round. The ABI GeneAmp^®^ type 9700 (ABI, Carlsbad, CA, USA) was used for the PCR, and the products were recovered by 2% agarose gel electrophoresis, purified by an AxyPrep DNA Gel Recovery Kit (AXYGEN) (Axygen Biosciences, Union City, CA, USA), eluted by Tris HCl, and quantified with the QuantiFluor™-ST (Promega, Madison, WI, USA). According to the Illumina MiSeq platform standard operating procedure, the purified amplified fragments were constructed into a library. Illumina’s MiSeq PE300 and MiSeq PE250 platforms were used for sequencing [55] (Table 1).

Processing of sequencing data. The raw 16S rRNA gene-sequencing reads were demultiplexed, quality-filtered via fastp version 0.20.0 [56], and merged via FLASH version 1.2.7 [57] according to the following criteria: (i) the 300 bp reads were truncated at any site with an average quality score of <20 over a 50 bp sliding window, truncated reads shorter than 50 bp were discarded, and reads containing ambiguous characters were also discarded; (ii) only overlapping sequences longer than 10 bp were assembled according to their overlapped sequence, the maximum mismatch ratio of overlapped regions was 0.2, and reads that could not be assembled were discarded; and (iii) samples were distinguished according to the barcode and primers, and the sequence direction was adjusted for exact barcode matching and 2 nucleotide primer mismatches.

Operational taxonomic units (OTUs) with a 97% similarity cut off were clustered using UPARSE (version 7.1, http://drive5.com/uparse/, accessed on 30 June 2022), and chimeric sequences were identified and removed. The taxonomy of each OTU representative sequence was analyzed via RDP Classifier (http://rdp.cme.msu.edu/, accessed on 30 June 2022) version 2.2 against the 16S rRNA database, using a confidence threshold of 0.7 [58].

Raw data were uploaded to the NCBI database for comparison. Raw data for rhizospheric bacterial and endophytic bacterial sequences were deposited in the NCBI Sequence Read Archive (SRA) database under accession number PRJNA856634 (accessed on 12 July 2022) and PRJNA856980 (accessed on 12 July 2022), respectively.

### 2.6. Statistical Analysis

Experimental data were recorded using Excel 2003 for ease of mathematical calculations. Statistical analysis was performed using SPSS statistics 22.0 (IBM Crop., Armonk, New York, NY, USA), and Duncan’s multiple range test was used to compare the means. Rhizosphere soil bacterial sequences, bacterial richness indexes (ACE and Chao1), bacterial diversity indexes (Shannon and Invsimpson), and the evenness index (Heip) were calculated in Mothur (version v.1.30.2 https://mothur.org/wiki/calculators/, accessed on 30 June 2022) [59]. Significance was based on 999 Monte Carlo permutations. Linear discriminant analysis (LDA) and an LDA effect size (LEfSe) method were used to identify significantly different bacterial communities in the different environmental samples [60]. The NetworkX tool kit (https://networkx.org/, accessed on 30 June 2022) was used for co-occurrence network analysis. PICRUSt gene function prediction: using information from the KEGG database, the KO, Pathway, and EC data were obtained, and the abundance of each functional category was calculated by OTU abundance. Online data analysis was performed using the free online platform of the Majorbio Cloud Platform (http://www.majorbio.com, accessed on 30 June 2022) from the Majorbio Bio-Pharm Technology Co., Ltd. (Shanghai, China).

## 3. Results

As seen in Table 2, the activities of soil *β*-glucosidase and phosphodiesterase in the rhizosphere of early and late-maturing pumpkin varieties were significantly higher than those of the background data (CK). The activity of aminopeptidase in the rhizospheres was also higher than that of the background data, but a significant difference was only seen in the rhizosphere of the early maturing varieties. Moreover, aminopeptidase activity in the early maturing rhizosphere was also significantly higher than that of the late-maturing variety (*p* < 0.05), suggesting that the early maturing varieties had a more abundant nitrogen supply in the rhizosphere during growth.

As shown in Table 3, the indexes of coverage exceeded 97%, indicating that the sequencing data were reasonable. Even though the indexes of soil bacteria diversity (Shannon and Invsimpson), richness (Ace and Chao1), and evenness (Heip) in the rhizospheres of the early and late-maturing pumpkins were all significantly higher than those of the background data (CK) (*p* < 0.05), there was no significant difference for each index between the early and late-maturing pumpkin varieties (Table 3).

Furthermore, to evaluate the extent of the similarity of the rhizospheric bacterial communities, a principal coordinate analysis (PCoA) at the OTU level was also performed (Figure 2). The results suggested that the rhizospheric bacterial compositions of both early and late-maturing pumpkins were quite similar (Figure 2a). The rhizospheric bacteria and CK clearly clustered into three and two taxa, respectively, and were distributed in both negative and positive directions of comp2 (count, map 2) based on a partial least squares discriminant analysis (PLS–DA). This result also indicated a different rhizosphere bacterial community structure for the two maturation varieties (Figure 2b). In other words, different soil bacteria were specifically recruited by early and late-maturing pumpkin varieties.

As shown in Figure 3a, the number of the soil-dominant bacterial phyla (relative abundances greater than 1%) in the background (CK), early (EM), and late (LM)-maturing varieties was 11, 9, and 9, respectively.

The dominant soil bacterial phyla in the CK were Proteobacteria (25.42%), Actinobacteriota (22.97%), Chloroflexi (18.54%), Acidobacteriota (11.44%), Firmicutes (7.81%), Bacteroidota (2.83%), Gemmatimonadota (2.66%), WPS-2 (1.79%), Myxococcota (1.48%), Patescibacteria (1.00%), and others (4.60%).

The dominant soil bacterial phyla in the rhizospheres of the EM varieties were Proteobacteria (27.12%), Actinobacteriota (23.73%), Chloroflexi (14.84%), Acidobacteriota (11.87%), Firmicutes (6.88%), Gemmatimonadota (3.86%), Bacteroidota (2.60%), Myxococcota (2.53%), and others (5.43%).

The dominant soil bacterial phyla in the rhizospheres of the LM varieties were Proteobacteria (27.08%), Actinobacteriota (24.87%), Chloroflexi (16.84%), Acidobacteriota (11.70%), Firmicutes (4.83%), Gemmatimonadota (2.8%), Bacteroidota (2.46%), Myxococcota (2.19%), and others (5.53%).

WPS-2 and Patescibacteria, two dominant bacterial phyla, were not detected in the EM and LM varieties when compared with the CK. Although the compositions of the dominant soil bacterial community in the rhizospheres of the EM and LM pumpkin varieties were similar to those of the CK, their proportions were different. The result indicated that the proportions of dominant soil bacteria in the rhizospheres were significantly different at the phylum level depending on the variety.

At the genus level, the number of dominant soil bacterial genera (i.e., those with relative abundances greater than 1%) among the CK, EM, and LM pumpkin varieties was 24, 22, and 21, respectively (Figure 3a).

First, *norank_f__norank_o__norank_c__AD3* (4.18%), *norank_f__norank_o__Gaiellales* (4.05%), *norank_f__Xanthobacteraceae* (49.79%), *norank_f__norank_o__norank_c__TK10* (3.27%), *norank_f__norank_o__Acidobacteriales* (2.98%), *Oceanobacillus* (2.59%), *Sphingomonas* (2.35%), *norank_f__norank_o__Elsterales* (2.33%), *norank_f__JG30-KF-CM45* (2.03%), *norank_f__norank_o__Vicinamibacterales* (1.93%), *norank_f__norank_o__norank_c__norank_p__WPS-2* (1.79%), *norank_f__Gemmatimonadaceae* (1.72%), *Acidothermus* (1.67%), *Gaiella* (1.59%), *Bacillus* (1.52%), *Conexibacter* (1.31%), *Bradyrhizobium* (1.29%), *norank_f__Roseiflexaceae* (1.22%), *unclassified_f_Rhizobiaceae* (1.10%), *norank_f__norank_o__IMCC26256* (1.03%), *Bryobacter* (1.03%), *Mycobacterium* (1.02%), *norank_f__67-14* (1.01%), and others (49.79%) were the dominant soil bacterial genera in CK.

Second, *norank_f__norank_o__Vicinamibacterales* (3.48%), *norank_f__norank_o__Gaiellales* (2.93%), *norank_f__Roseiflexaceae* (2.86%), *norank_f__Gemmatimonadaceae* (2.75%), *norank_f__Vicinamibacteraceae* (2.44%), *norank_f__Xanthobacteraceae* (2.35%), *Sphingomonas* (2.32%), *norank_f__norank_o__norank_c__TK10* (2.20%), *Bacillus* (2.17%), *norank_f__JG30-KF-CM45* (2.07%), *Gaiella* (1.82%), *Arthrobacter* (1.79%), *Oceanobacillus* (1.71%), *norank_f__SC-I-84* (1.50%), *norank_f__67-14* (1.49%), *1norank_f__norank_o__C0119* (1.20%), *Bradyrhizobium* (1.19%), *norank_f__norank_o__IMCC26256* (1.15%), *norank_f__norank_o__norank_c__KD4-96* (1.06%), *Bryobacter* (1.02%), *Mycobacterium* (1.01%), and others (54.41%) were the dominant soil bacterial genera in the EM rhizospheres.

By contrast, *norank_f__Xanthobacteraceae* (3.22%), *norank_f__norank_o__Gaiellales* (3.00%), *norank_f__Roseiflexaceae* (2.50%), *norank_f__norank_o__norank_c__TK10* (2.44%), *Sphingomonas* (2.06%), *norank_f__Gemmatimonadaceae* (1.83%), *norank_f__norank_o__norank_c__AD3* (1.82%), *norank_f__norank_o__Elsterales* (1.75%), *norank_f__Vicinamibacteraceae* (1.72%), *Arthrobacter* (1.60%), *norank_f__JG30-KF-CM45* (1.58%), *Oceanobacillus* (1.49%), *norank_f__norank_o__Acidobacteriales* (1.35%), *Bacillus* (1.34%), *Bradyrhizobium* (1.21%), *norank_f__norank_o__IMCC26256* (1.20%), *Gaiella* (1.16%), *Conexibacter* (1.14%), *norank_f__norank_o__norank_c__KD4-96* (1.11%), *norank_f__SC-I-84* (1.04%), and others (54.32%) were the dominant soil bacterial genera in the LM rhizospheres.

Meanwhile, *norank_f__norank_o__Vicinamibacterales*, *norank_f__67-14*, *norank_f__norank_o__C0119*, *Bryobacter*, and *Mycobacterium* were the special dominant soil bacteria in the rhizospheres of the EM varieties. By contrast, *norank_f__norank_o__norank*_c__AD3, *norank_f__norank_o__Acidobacteriales*, *norank_f__norank_o__Elsterales,* and *Conexibacter* were unique dominant soil bacteria in the LM rhizospheres (Figure 3b).

At the genus level, the number of unique soil bacterial genera associated with the CK and the EM and LM varieties was 18, 41, and 28, respectively, and the total number of corresponding soil bacterial genera was 889, 1069, and 1053 (Figure 4a).

Moreover, 5146, 7376, and 7548 bacterial OTUs were found in the CK, EM, and LM soil, respectively. Among all three, 4326 common soil-bacterial OTUs were detected in the rhizospheres, and 199, 536, and 575 unique bacterial OTUs were detected in the CK, EM, and LM soil, respectively (Figure 4b).

Furthermore, in comparison with the late-maturing varieties, *Pedomicrobium*, *Nordella*, *MND1*, unclassified_f__*Comamonadaceae Noviherbaspirillum*, norank_f__*TRA3-20*, *Sporosarcina*, unclassified_f__*Chitinophagaceae*, norank_f__*Chitinophagaceae*, and norank_f__*Pedosphaeraceae* were enriched in the rhizospheres of the early maturing varieties. In contrast, only *Amycolatopsis*, *Pseudonocardia*, and *Oligoflexus* were enriched in the rhizospheres of the LM varieties (Figure 5).

The top 30 rhizospheric bacteria were selected to construct a single-factor correlation network analysis (*p* < 0.01). The top-10 rhizospheric bacteria with the strongest correlation with other rhizospheric bacteria were *unclassified_f__Rhizobiaceae*, *norank_f__Xanthobacteraceae*, *norank_f__JG30-KF-CM45*, *Conexibacter*, *norank_f__norank_o__Elsterales*, *Bryobacter*, *norank_f__norank_o__norank_c__AD3*, *norank_f__norank_o__Acidobacteriales*, *norank_f__norank_o__norank_c__TK10*, and *Bradyrhizobium.* Among them, only *norank_f__JG30-KF-CM45* correlated negatively with other bacteria (Figure 6).

The functional profiles of soil bacteria in the EM and LM rhizospheres were quite similar. They all consisted of functions that are listed A–Z in Figure 7. The top-10 relative abundance ratios of soil-bacterial PICRUSt functions from largest to smallest were amino acid transport and metabolism, general function prediction only, energy production and conversion, transcription, carbohydrate transport and metabolism, cell wall/membrane/envelope biogenesis, replication, recombination and repair, inorganic ion transport and metabolism, and signal transduction mechanisms.

The relative abundances of genes involved in RNA processing and modification, cell-cycle control, cell division, chromosome partitioning, translation, ribosomal structure and biogenesis, and cell motility were higher in EM pumpkins than in LM pumpkins, whereas those of the other genes were lower (Figure 7).

In addition, the Shannon, Invsimpson Ace, Chao1, and Heip indices, which describe the endophytic bacterial diversity, richness, and evenness, showed no significant differences between early and late-maturing pumpkin varieties (Table 4).

To evaluate the similarity of the endophytic bacterial communities, principal coordinate analysis (PCoA) at the OTU level was also performed (Figure 8). The results suggested a relative similarity in bacterial compositions for both EM and LM pumpkins (Figure 8a). Moreover, the endophytic bacteria clearly clustered into two taxa and were distributed in negative and positive directions of comp1 (count, map 1) depending on the partial least squares discriminant analysis (PLS–DA). This result also indicated that the endophytic bacterial community structure was significantly different between the EM and LM pumpkins (Figure 8b).

The number of the dominant endophytic bacterial phyla (i.e., relative abundances greater than 1%) between the EM and LM varieties was 6 and 5, respectively (Figure 9a).

First, Actinobacteriota (58.94%), Proteobacteria (33.43%), Firmicutes (1.88%), Bacteroidota (1.64%), Gemmatimonadota (1.05%), and others (2.12%) were the dominant endophytic bacterial phyla of the EM varieties. By contrast, Actinobacteriota (50.19%), Proteobacteria (41.24%), Abditibacteriota (3.19%), Bacteroidota (1.63%), and others (2.13%) were dominant among the LM varieties.

Firmicutes and Gemmatimonadota were the unique dominant endophytic bacterial phyla in the EM varieties, whereas Abditibacteriota was the unique endophytic bacterial phylum in the LM varieties.

In addition, at the genus level, the numbers of dominant endophytic bacteria—those with relative abundances greater than 1%—were 20 and 21 in the LM and EM varieties, respectively (Figure 9b).

The dominant endophytic bacterial genera in LM were *Nocardioides* (22.67%), *Quadrisphaera* (7.53%), *Marmoricola* (7.42%), *Actinomycetospora* (7.05%), *Methylobacterium-Methylorubrum* (6.40%), *Sphingomonas* (4.96%), *Pseudokineococcus* (4.72%), *Aureimonas* (4.54%), *Klenkia* (4.52%), *Microbacterium* (3.90%), *Devosia* (3.21%), *Allorhizobium-Neorhizobium-Pararhizobium-Rhizobium* (2.00%), *Pseudonocardia* (1.70%), *Kineococcus* (1.51%), *Kineosporia* (1.44%), *unclassified_f__Rhizobiaceae* (1.40%), *Rhodococcus* (1.25%), *Skermanella* (1.17%), *Massilia* (1.09%), and others (20.28%).

In the EM, these were *Ralstonia* (10.78%), *Nocardioides* (9.87%), *Marmoricola* (6.71%), *Sphingomonas* (5.34%), *Microbacterium* (4.98%), *Actinomycetospora* (4.48%), *Devosia* (4.42%), *Quadrisphaera* (4.31%), *Methylobacterium-Methylorubrum* (4.08%), *Aureimonas* (3.44%), *Pseudokineococcus* (3.32%), *Abditibacterium* (3.19%), *Klenkia* (2.65%), *Pseudonocardia* (2.51%), *Kineococcus* (2.48%), *unclassified_f__Rhizobiaceae* (1.85%), *Kineosporia* (1.84%), *Allorhizobiu*m-*Neorhizobium-Pararhizobium-Rhizobium* (1.22%), *unclassified_f__Solirubrobacteraceae* (1.21%), *Brevundimonas* (1.12%), and others (19.18%).

Among these, *Rhodococcus*, *Skermanella*, and *Massilia* were the unique endophytic bacterial genera of the LM pumpkin varieties, whereas for the LM varieties, they were *Ralstonia*, *Abditibacterium*, *unclassified_f__Solirubrobacteraceae*, and *Brevundimonas*.

Furthermore, the number of unique endophytic bacterial genera in the early and late-maturing pumpkins was 54 and 30, respectively, and the total number of endophytic bacterial genera was 349 and 325, respectively (Figure 10a).

In total, 660 common endophytic bacterial OTUs were found, and 145 and 99 unique OTUs were detected in the early and late-maturing pumpkins, respectively. The results suggested that the number of unique and total endophytic bacterial genera and OTUs of the EM varieties was higher (Figure 10b).

Moreover, the number of dominant EM endophytic bacterial groups was similar to that in the LM varieties. For example, *Candidatus_Captivus* and *Noviherbaspirillum* norank_f__*Ilumatobacteraceae* were enriched in EM endophytic bacteria, whereas norank_f__*Rhizobiaceae*, *Pelagibacterium*, *Novosphingobium*, *Mariniflexile*, and *Kineosporia* were enriched in the LM varieties (Figure 11).

Furthermore, Nakamurella, Nocardioides, Kineococcus, Marmoricola, Actinomycetospora, Rhodococcus, Truepera, unclassified_f__Rhizobiaceae, Longimicrobium, and Pseudokineococcus were the top-10 endophytic bacteria with the strongest correlation with other endophytic bacteria. Most of them correlated positively, but Methylobacterium–Methylorubrum showed a negative correlation with Truepera (Figure 12).

In addition, the functions of 24 endophytic bacteria in the EM pumpkins—translation, ribosomal structure and biogenesis, cell wall/membrane/envelope biogenesis, replication, recombination and repair, post-translational modification, protein turnover, chaperones, intracellular trafficking, secretion, vesicular transport, and cell motility—were all relatively higher (Figure 13).

## 4. Discussion

Investigation of the soil and endophytic microbial community structure can provide new insight into microorganisms in the rhizospheric and endophytic microbiomes that might promote plant growth [61]. In previous studies, we found that soil bacteria and endophytic bacteria play a prominent role in the maturation of tobacco and rice seeds [62,63]. Early and late-maturing pumpkin varieties were simultaneously planted and grown in the same field under identical management, but they blossomed and fruited at different times.

### 4.1. Effects of Early and Late-Maturing Pumpkin Varieties on Soil Enzymes

It is well-known that soil enzymes are used as bioindicators for soil fertility and are strongly related to microorganisms [30]. For instance, *β*-glucosidase is widely distributed in nature and is related to the carbon cycle, thus it can be considered a soil quality indicator based on its direct relationship with the quantity and quality of soil organic matter [64]. The activities of phosphatase are closely correlated with the organic matter content [65] and depend on factors such as soil properties, organism interactions, and plant cover [66]. Aminopeptidase, which is related to the N-cycle, is very sensitive to environmental change [67]. Our results revealed that the activities of *β*-glucosidase and phosphatase in the rhizospheres showed no significant differences between the EM and LM varieties. However, the activity of aminopeptidase in the EM rhizosphere was significantly higher, indicating a faster N-cycle.

### 4.2. Effects of Early and Late-Maturing Pumpkin Varieties on Rhizosphere and Endophytic Bacteria Communities

In addition, early or late maturation is a complex physiological and biochemical reaction that involves the regulation of a variety of hormones. Among them, exogenous auxin is considered to be a negative regulator of fruit ripening [68,69,70]. Plant endogenous auxin and cytokinin promote the formation of pumpkin caryopsis. Gibberellins plays an important role in promoting cell elongation and cell wall extensibility [71]; therefore, a decrease in ethylene promotes fruit development [72]. The amount of abscisic acid increased rapidly during fruit development [68] and correlated positively with actinomycetes and negatively with proteobacteria [73]. Bacteroidetes are related to the degradation of various organic substances [74] such as *Rhodococcus* and *Ralstonia*, which produce auxin [75] and ethylene [76], respectively. Furthermore, previous studies confirmed that *Bacillus* induced the accumulation of abscisic acid [77] and gibberellin [78]; *Arthrobacter* promoted plant growth [79] and converted atmospheric nitrogen for plant use [80]; and *Bradyrhizobium* produced auxin [81].

Our results revealed that the proportions of Proteobacteria, Actinobacteriota, Bacteroidetes, and Firmicutes in the EM rhizospheres were higher compared with the LM ones. Additionally, Firmicutes and Gemmatimonadota were the unique, dominant, endophytic bacterial phyla. Furthermore, the unique, dominant, endophytic bacterial genus was *Rhodococcus*. In contrast, *Ralstonia* was the unique, dominant, endophytic bacterial genus of the LM varieties.

*Bacillus* and *Arthrobacter* were the dominant soil bacterial genera in the EM rhizospheres, and their abundance was higher than that of the LM varieties. The abundance of *Bradyrhizobium* in the LM varieties was higher compared with EM varieties.

Therefore, EM pumpkin varieties can be concluded to have a higher amount of abscisic acid and gibberellin, whereas a higher amount of auxin can be inferred in LM varieties according to the enrichment of rhizospheric and endophytic bacteria.

PICRUSt function prediction analysis also revealed that soil bacterial functions—such as RNA processing and modification, cell-cycle control, cell division, chromosome partitioning, translation, ribosomal structure and biogenesis, and cell motility in the rhizosphere—were all higher in the EM varieties, which indicated that soil bacteria in the EM rhizosphere had stronger metabolic functioning. However, the functions of endophytic bacteria—amino acid transport and metabolism and energy production—were similar between the two pumpkin varieties.

## 5. Conclusions

The early maturing varieties had a faster N-cycle, and the enrichment of abscisic-acid- and gibberellin-producing bacteria in the rhizospheres or endophytes may be important for early ripening. Moreover, *Rhodococcus*, *Bacillus*, and *Arthrobacter* can be considered as the functional bacteria for promoting ripening. On the other hand, *Ralstonia* is thought to be the functional bacterium that delays ripening.

## Figures and Tables

**Figure 1 microorganisms-10-01667-f001:**
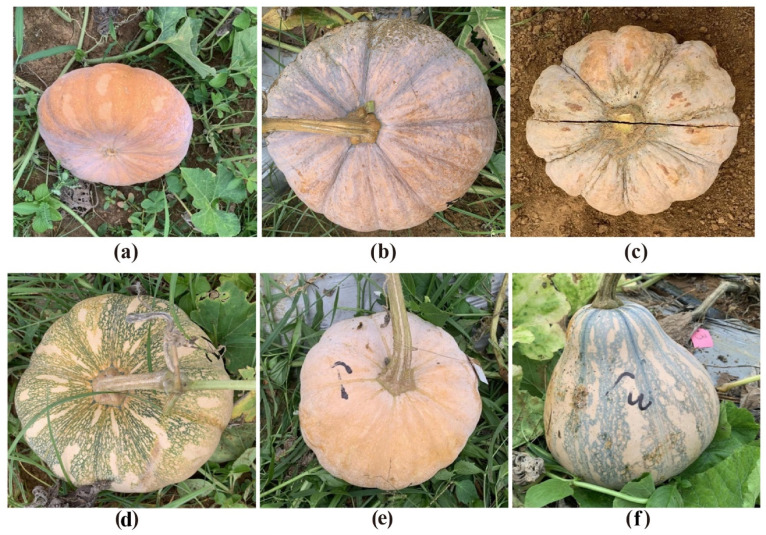
The appearance and morphological characteristics of early and late-maturing pumpkin varieties. (**a**) G1510-252-3-6; (**b**) G1681-2; (**c**) G1536-169-3-11; (**d**) G1601-3-2; (**e**) G1713-2; (**f**) G1215-1.

**Figure 2 microorganisms-10-01667-f002:**
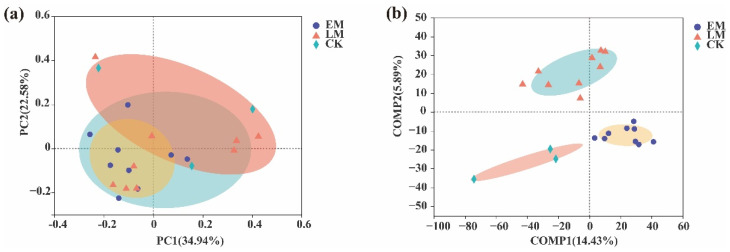
Comparison of soil bacteria in the rhizosphere between early (EM) and late (LM)-maturing pumpkin varieties. (**a**) PCoA of soil bacterial communities at the OTU level. (**b**) PLS-DA score plot of soil bacterial communities in the rhizosphere between the early and late-maturing varieties. Shaded parts are grouped circles.

**Figure 3 microorganisms-10-01667-f003:**
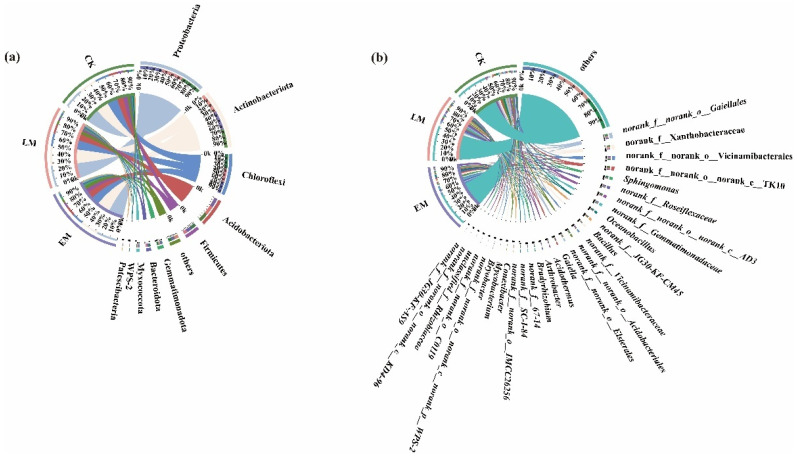
Distribution of dominant soil bacteria in the rhizosphere of CK and the EM and LM pumpkin varieties at the (**a**) phylum and (**b**) genus levels.

**Figure 4 microorganisms-10-01667-f004:**
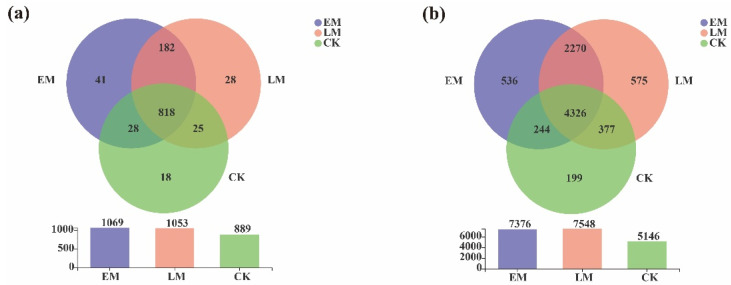
Venn diagrams of the dominant soil bacteria in the rhizosphere at the genus (**a**) and OTU (**b**) levels among the background (CK), early maturing (EM), and late-maturing (LM) pumpkins.

**Figure 5 microorganisms-10-01667-f005:**
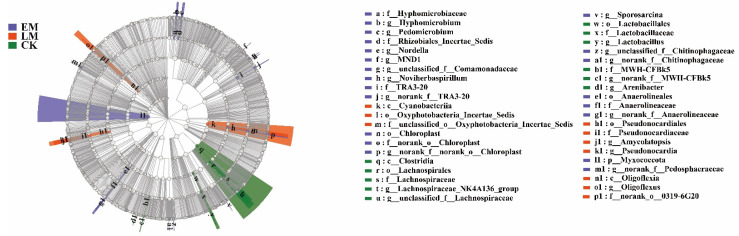
LEfSe analysis of soil bacteria in the rhizosphere among the background (CK) and the early maturing (EM) and late-maturing (LM) pumpkins. Different colour regions represent different constituents (Purple: EM; Red, LM; Green: CK). Circles indicate phylogenetic level from phylum to genus. The diameter of each circle is proportional to the abundance of the group. Different prefixes indicate different levels (p: phylum; c: class, o: Order; f: Family; g: Genus).

**Figure 6 microorganisms-10-01667-f006:**
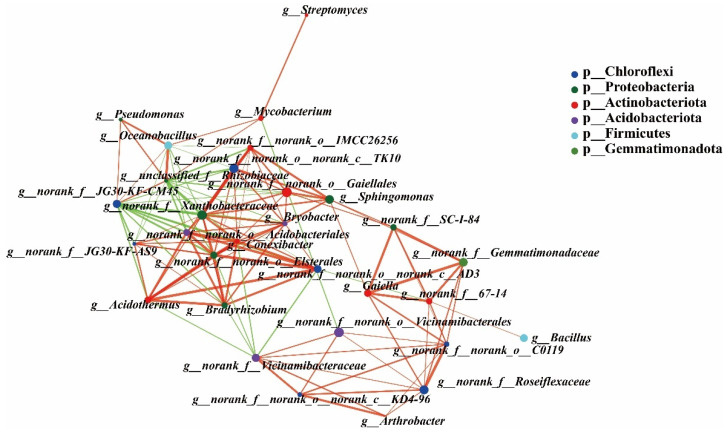
Co-occurrence network analysis of rhizosphere bacteria among the CK and the EM and LM pumpkin varieties. The size of the node is proportional to the genera abundance. Node color corresponds to phylum taxonomic classification, and marked nodes represent significant differences. Edge color represents positive (green) and negative (red) correlations, and the edge thickness is equivalent to the correlation values, *p* < 0.01.

**Figure 7 microorganisms-10-01667-f007:**
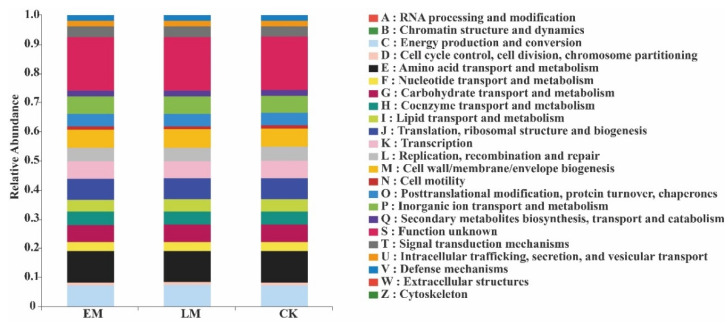
Relative abundance of PICRUSt-inferred functions of soil bacteria in the rhizosphere of EM and LM pumpkin varieties and the CK.

**Figure 8 microorganisms-10-01667-f008:**
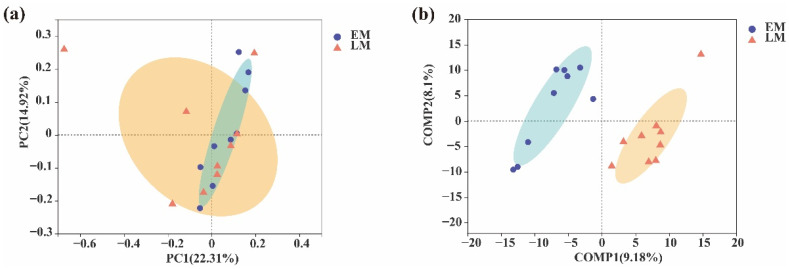
Comparison of endophytic bacteria between EM and LM pumpkin varieties. (**a**) PCoA of endophytic bacterial communities at the OTU level. (**b**) PLS-DA score plot of endophytic bacterial communities. Shaded parts are grouped circles.

**Figure 9 microorganisms-10-01667-f009:**
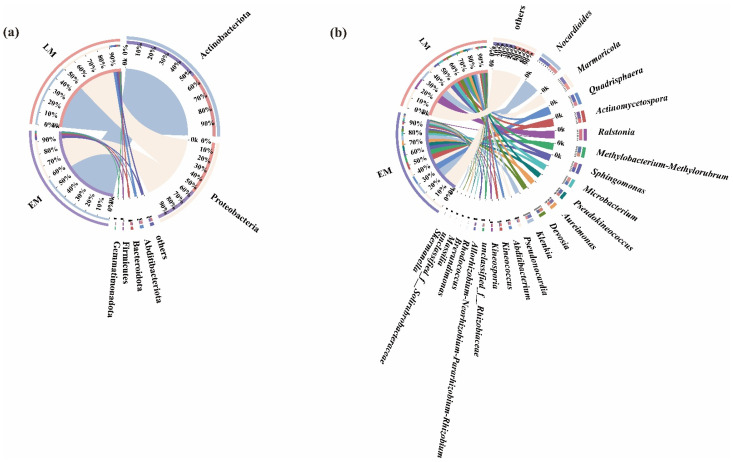
Distribution of endophytic bacteria between EM and LM pumpkin varieties at the (**a**) phylum and (**b**) genus levels.

**Figure 10 microorganisms-10-01667-f010:**
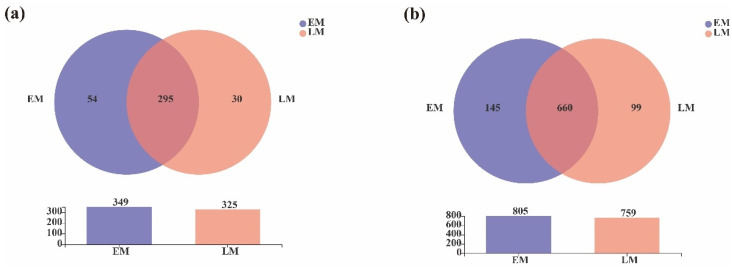
Venn diagrams of the endophytic bacteria at the (**a**) genus and (**b**) OTU levels between EM and LM pumpkin varieties.

**Figure 11 microorganisms-10-01667-f011:**
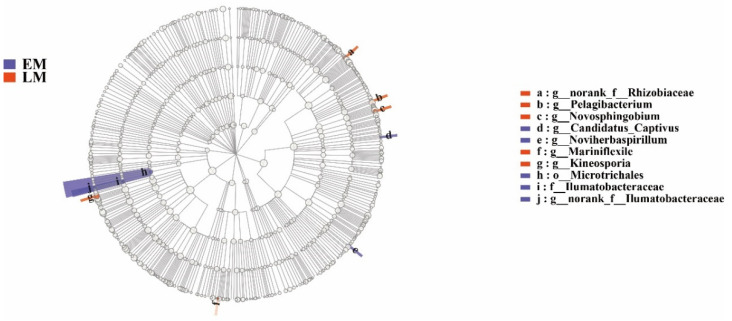
LEfSE analysis of endophytic bacteria between the early maturing (EM) and late-maturing (LM) pumpkin varieties.

**Figure 12 microorganisms-10-01667-f012:**
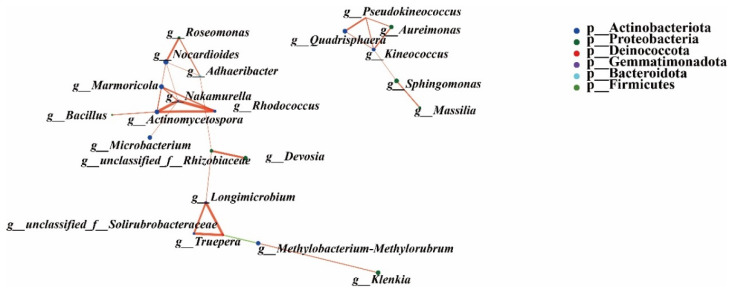
Co-occurrence network analysis of endophytic bacteria between the EM and LM pumpkin varieties. The red line indicates a positive interaction; the green line indicates a negative interaction; and marked nodes represent significant differences, *p* < 0.01.

**Figure 13 microorganisms-10-01667-f013:**
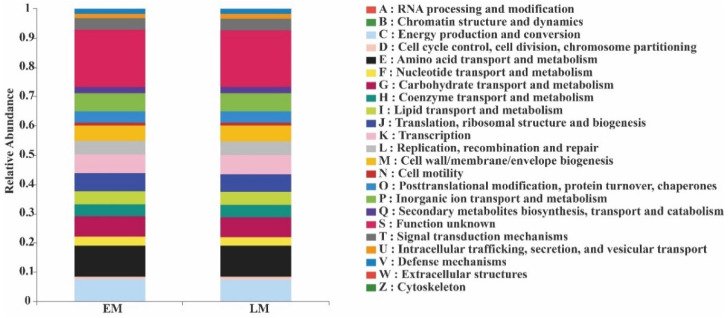
Relative abundance of PICRUSt-inferred endophytic bacterial functions between EM and LM pumpkin varieties.

**Table 1 microorganisms-10-01667-t001:** Sequencing type and primer sequence.

Sequencing Type	Primer Name	Primer Sequence	Length	Sequencing Platform
Rhizospheric bacterial	338F	5′-ACTCCTACGGGAGGCAGCAG-3′	311 bp	MiSeq PE300
	806R	5′-GGACTACHVGGGTWTCTAAT-3′		
Endophytic bacterial	799F	5′-AACMGGATTAGATACCCKG-3′	394 bp	MiSeq PE250
	1193R	5′-ACGTCATCCCCACCTTCC-3′		

**Table 2 microorganisms-10-01667-t002:** Soil enzyme (nmol g^−1^ min^−1^ at 30 °C) activities in the early and late-maturing pumpkin varieties.

Sample	*β*-Glucosidase	Phosphatase	Aminopeptidase
EM	0.27 ± 0.13 ^a^	2.83 ± 1.11 ^a^	14.18 ± 1.99 ^a^
LM	2.44 ± 0.69 ^a^	2.44 ± 0.69 ^a^	10.81 ± 2.47 ^b^
CK	0.12 ± 0.06 ^b^	1.14 ± 0.26 ^b^	9.25 ± 0.67 ^b^

Note: All data are presented as means ± SD (standard deviation). Groups compared using one-way ANOVA. Different letters in the same column indicate significant differences among treatments at *p* < 0.05.

**Table 3 microorganisms-10-01667-t003:** Soil bacterial diversity in the early (EM) and late-maturing (LM) pumpkins.

	Shannon	Invsimpson	Ace	Chao1	Heip	Coverage
EM	6.78 ± 0.17 ^a^	312.00 ± 61.16 ^a^	4807.82 ± 464.85 ^a^	4782.72 ± 433.63 ^a^	0.26 ± 0.02 ^a^	0.97
LM	6.71 ± 0.15 ^a^	286.85 ± 64.78 ^a^	4711.95 ± 134.07 ^a^	4653.41 ± 102.38 ^a^	0.25 ± 0.03 ^a^	0.97
CK	6.36 ± 0.12 ^b^	180.15 ± 26.42 ^b^	4058.39 ± 604.8 ^a^	4036.71 ± 520.56 ^a^	0.19 ± 0.01 ^b^	0.98

Note. All data are presented as the means ± SD (standard deviation). The Student’s t-test was performed (*p* < 0.05). Different letters in the same column indicate significant differences among treatments at *p* < 0.05.

**Table 4 microorganisms-10-01667-t004:** Endophytic bacterial diversity in the early (EM) and late-maturing (LM) pumpkins.

	Shannon	Invsimpson	Ace	Chao1	Heip	Coverage
EM	4.27 ± 0.27 ^a^	31.96 ± 12.69 ^a^	455.7 ± 92.83 ^a^	453.5 ± 90.16 ^a^	0.2 ± 0.06 ^a^	0.99
LM	3.95 ± 1.00 ^a^	30.07 ± 15.91 ^a^	479.5 ± 86.69 ^a^	478.73 ± 92.24 ^a^	0.17 ± 0.07 ^a^	0.99

Note. All data are presented as the means ± SD (standard deviation). The Student’s t-test was performed (*p* < 0.05). Different letters in the same column indicate significant differences among treatments at *p* < 0.05.

## Data Availability

Raw data for rhizospheric bacterial and endophytic bacterial sequencing were deposited in the NCBI Sequence Read Archive (SRA) database under accession number PRJNA856634 (accessed on 12 July 2022) and PRJNA856980 (accessed on 12 July 2022), respectively.

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
