# Peer review of "A Comparison of Rhizospheric and Endophytic Bacteria in Early and Late-Maturing Pumpkin Varieties"

_microorganisms, 2022, doi:10.3390/microorganisms10081667_

Round 1

Reviewer 1 Report

Chen et al.2022. A Comparison of Rhizospheric and Endophytic Bacteria in Early and Late-Maturing Pumpkin Varieties

 Summary

This study analyzes the possible contribution of rhizosphere/endophytic bacteria in the maturation of pumpkins. The authors looked for differences between the structure/function of bacterial communities of pumpkin varieties with different maturity and their relationship with enzyme activities related to functional changes in the rhizosphere. Authors found differences in the abundance of putative phytohormone-producing bacteria, suggesting a role in pumpkin maturation.

Although the MS could be a valuable contribution, crucial aspects need further description or clarification in materials and methods to support the results and conclusions. Therefore I could not recommend it for publication in its current version. See comments below.

 Introduction.

This section should address how 'fertility parameters', particularly enzymatic activities, could be related to ripening.

L44. Inappropriate citation. Why is erythromycin a plant hormone? Verify this statement; note that this is not described in the study by Tudzynski, et al., 1998.

L54-62. It would make more sense to review these topics in the context of the rhizosphere. If that was already the focus, indicate it more explicitly.

L56. Indicate what types of enzymes?

L57 The statement needs more literature support; please elaborate. Notice that the reference by Kutílek et al. 2006 (37) only supports the soil's physical properties, particularly micromorphology.

L64. I suggest using 'microhabitat' instead of 'microecosystem.'

 Materials and methods

-It is unclear whether endophytes were obtained from roots (L107-108) or stems (L117). In the first case, what procedures were carried out to differentiate endophytic bacteria from those colonizing the rhizosphere or rhizoplane.

 I notice that this MS and methodological approaches (L137-150) are pretty similar to those recently reported by Xia et al. 2022 (see below), where the current corresponding author has participated. I would suggest rephrasing sections of M&M similar to that work or citing it.

Xiao, J., Chen, S. Y., Sun, Y., Yang, S. D., & He, Y. (2022). Differences of rhizospheric and endophytic bacteria are recruited by different watermelon phenotypes relating to rind colors formation. Scientific Reports, 12(1), 1-13.

 L98. Missing information. What was the phenological stage of the pumpkin during sample collection?

L107. What happened with these roots? They are not described later in M&M.

L111. Physicochemical properties of soils are missing in the results. Please verify.

L112. The activity of β-Glucosidase needs to be addressed first in the introduction; why beta-glucosidase? Aminopeptidase activity (Same comment)

L125-128. Describe how endophytic bacterial DNA was obtained; this is a critical part of the study. In addition, add a proper citation for primers used for endophytes.

 L 132. The sentence 'Blue Fluorescence quantitative system' seems out of context, revise.

 L160. Statistical Analysis. Rhizosphere soil bacterial sequences..' and endophytic'?

L156-159. Statistical Analysis. Details for co-occurrence network analysis are missing; please add.

L164. Revise, fungal communities are not part of this study.

 Results

I suggest organizing the results by subsections; it is hard to see the change from one topic to another.

L 176. If enzymatic activity is an indicator of fertility and higher availability of nutrients, would it not be more accurate to quantify N in plants? Or, at least in the rhizosphere? Same comment for L412. I understand enzyme activities in this study indicate functional changes in rhizosphere microbial communities.

L286-297. Although interesting, the purpose of network analysis between rhizosphere and endophytic bacteria is unclear. Later in L380-384, the same analysis correlates only endophytic bacteria. Please explain the idea of these analyses.

L318. 'endophytic bacteria clustered in two taxa', please clarify.

L351-357 Italics in all genera.

 Discussion

L403-404. See comment for L176 and L412.

417. Indicate the meaning of GA.

Conclusions

L446. Nitrogen content was not addressed in this stud; please rephrase.

 Table 1. Footnote: Indicate what type of test was performed to test soil enzyme statistical differences and size of n.

Fig. 2. Consistency. Use consistently CK, not BG.

Reviewer 2 Report

In general, the authors have done a good job explaining the background information necessary to appreciate the rationale and results of the experiments. The manuscript was prepared correctly. Methodology and analysis of results rather don't raise any objections.
However, some minor amendments are needed. The discussion of the results is written a bit generally. There are papers related to this topic that the authors did not cite. An assessment putting the findings into perspective and make a solid conclusion is missing. The authors should emphasize more the novelty and usefulness of the results.

Round 2

Reviewer 1 Report

Dear Editor,

The authors addressed all my comments and substantially improved the manuscript; however, one point still needs attention. The purpose of co-occurrence networks between endophytic and rhizosphere bacteria is still unclear. I encourage the authors to include in the MS the biological meaning of these correlations and discuss them (especially those taxa listed in lines 289-295); what was their relevance to the whole network?

Minor detail

L426, Although an increase in gibberellins is the most plausible explanation, they were not quantified in this study; please, rephrase this sentence to avoid speculation

Author Response

Dear Editors and Reviewers:

Thank you for your letter and the reviewers’ comments concerning our manuscript entitled “A Comparison of Rhizospheric and Endophytic Bacteria in Early and Late-Maturing Pumpkin Varieties” (Submission ID: microorganisms-1837716). Those comments are all valuable and very helpful for revising and improving our paper, as well as the important guiding significance tour researches. We have read comments carefully and have made correction which we hope meet with approval. Revised portion are marked in Red in the paper. The main corrections in the paper and the responds to the reviewer’s comments are as flowing:

Response to Reviewer 1 Comments

The authors addressed all my comments and substantially improved the manuscript; however, one point still needs attention. The purpose of co-occurrence networks between endophytic and rhizosphere bacteria is still unclear. I encourage the authors to include in the MS the biological meaning of these correlations and discuss them (especially those taxa listed in lines 289-295); what was their relevance to the whole network?

Response 1: Thank you so much for your valuble comments here. As the pumpkin ripening was found in relating to rhizospheric and endophytic bacteria, meanwhile, what is the relationship between the rhizospheric and the endophytic bacteria? Whether the shaping of the endophytic bacterial community was related to the rhizospheric bacteria. Therefore, the purpose of co-occurrence networks between endophytic and rhizospheric bacteria was to check whether the endophytic bacteria shaping were also affected by rhizospheric bacteria. Moreover, If this answer is still not good enough for explanation, is it necessary to delate this part in the manuscript?  

Minor detail

L426, Although an increase in gibberellins is the most plausible explanation, they were not quantified in this study; please, rephrase this sentence to avoid speculation

Response 2: The authors have rephased the sentence in manucript according to the Reviewer’s suggestion (L445-448).

Special thanks to you for your good comments.
